



# A global blockage parametrization for engineering wake models

Niels Goedegebure[1], Mikko Folkersma[2], and Jakob Maljaars[3]

[1,2,3]Whiffle Precision Weather Forecasting BV, The Netherlands

**Correspondence:** Mikko Folkersma (mikko.folkersma@whiffle.nl)

**Abstract.** Whereas engineering wake models can be used to efficiently provide energy production estimates for wind turbine sites, recent studies indicate the importance of a *global* blockage effect becomes manifest for larger assets. This global blockage effect is caused by site-scale interactions with the atmospheric boundary layer, and results in a wind speed deficit upstream of the asset. This paper presents an efficient and accurate parametrized global blockage model which integrates into existing engineering wake models. The central idea behind this global blockage model is to interpret the wind farm site as a parametrized porous object, subjected to an ambient flow field. We calibrate and benchmark our model through high-resolution LES model data for a representative offshore site using a calibrated wake deficit shape parameter. Results show significant improvements in turbine-level energy production prediction accuracy when compared to results obtained without any blockage model and results obtained with the local *self-similar* blockage model. The parametrized global blockage model has a significantly lower computational footprint compared to local blockage models. We conclude that not taking (global) blockage into account sufficiently can yield a tendency to overestimate the strength of the turbine wake deficit effects when calibrating wake deficit shape parameters. Finally, we note that the spatial distribution of (global) blockage and wake deficit errors can easily lead to error cancellation when aggregating over binned wind directions.

## 1 Introduction

Engineering wake models provide an efficient method to estimate energy production for wind assets, resulting in approximations with low computational effort. Two common implementations include FLORIS (NREL, 2024) and PyWake (Pedersen et al., 2023). Engineering wake models consist of a *wake deficit* model, modeling the shape and intensity of wakes created by turbines and a *superposition model* computing the deficit for overlapping wakes. These models are combined with turbine and site specifications and are run using a numerical solver, yielding an energy production estimate for steady-state flow given the pre-specified wind conditions. To increase the accuracy of these methods, several calibration approaches have been proposed, by e.g. using wind tunnel data (Campagnolo et al., 2022), real world wind farm measurement data (Liu et al., 2018; Keim, 2024; Teng and Markfort, 2020; van Binsbergen et al., 2024) and model data results from computational fluid dynamics (CFD) simulations, such as Large Eddy Simulation (LES) or Reynolds-averaged Navier–Stokes (RANS) results (Cathelain et al., 2020).

Furthermore, depending on the particular wake model solver, a *blockage model* may be included, resulting in a deceleration of turbine upstream wind speeds. Recently, blockage effects and the modeling thereof has gained the attention of the



community, since neglecting this effect can lead to a strong positive bias of wind energy production (Bleeg et al., 2018). Engineering wake blockage models typically only consider a *local* blockage induction zone per turbine. Approaches based upon this strategy include the *self-similar* model of Troldborg and Meyer Forsting (2017), the *rankine half-body* model of Gribben

and Hawkes (2019), the wake and blockage model setup of Nygaard et al. (2020) and the *vortex-cylinder* model of Branlard and Gaunaa (2014). Model comparisons generally show relatively similar results for deficit fields of different blockage model formulations when assessed at turbine-level (Branlard et al., 2020; Çam, 2022).

However, the underlying assumption that blockage effects are a superposition of the effect at individual turbines is under increasing scrutiny as research shows a *global blockage* (GB) effect plays an important role in big assets (Meyer Forsting

et al., 2023). Superposition of local blockage effects typically yields an underprediction of the total blockage wind speed velocity deficit (Allaerts and Meyers, 2019; Sebastiani et al., 2022; Stipa et al., 2024). Recent research points towards global blockage emerging from effects at wind farm scale interacting with the atmospheric boundary layer (Maheshwari et al., 2024; Schneemann et al., 2021). This hence provides a motivation for using specific large-scale global (i.e. wind farm scale) blockage models in engineering wake models instead of the superposition of local blockage effects.

A number of global blockage models for wake models have been proposed in literature. In the three-layer model of Allaerts and Meyers (2019), the meso-scale "gravity wave" induced power loss effect is modeled by a flow model that includes the vertical displacement caused by the gravity wave effect as detailed by Smith (2010). This approach shows improved results and was recently expanded further into the meso-micro approach of Devesse et al. (2024) and the multi-Scale Coupled Model of Stipa et al. (2024). The results are compared to accurate LES results and show a significant improvement in accuracy compared

to the three-layer model and wake-only setups.

The aforementioned models, however, introduce multiple new model stages and require more extensive input parameter information compared to existing engineering wake models. While parametrized global blockage approaches have been proposed such as by Centurelli et al. (2021), as of current no thoroughly-tested model appears to fill the gap for an effective reduced-order parametrized global blockage model which easily integrates into existing engineering wake models.

In this paper, we propose such a parametrized global blockage model that can be used as an extension on top of existing engineering wake models. The model is based on a porous-media-like approach using the atmospheric boundary layer height and the turbine thrust coefficient. Benchmarking and calibration is carried out on a large, fictitious offshore site and shows an improved correspondence with high fidelity LES-generated energy production data when compared to existing wake-only and local blockage approaches. Furthermore, test results highlight the shortcomings of not including a global blockage model for

wake deficit model parameter calibration and the risk of error cancelation for long-term site energy production assessments.

The structure of this paper is as follows. In Section 2, the new model on global blockage is presented, discussing the assumptions and model formulations. In Section 3, a description of the components of the engineering wake models used in this study is provided, including an implementation of the parametrized global blockage model. The wake model with different blockage models is then applied to a case study on the planned offshore *IJmuiden Ver* wind asset, of which results are discussed

in Section 5. Section 6 draws conclusions and discusses the advantages and shortcomings of the model, including pathways for future research and improvements.



## 2 Parametrized global blockage model formulation

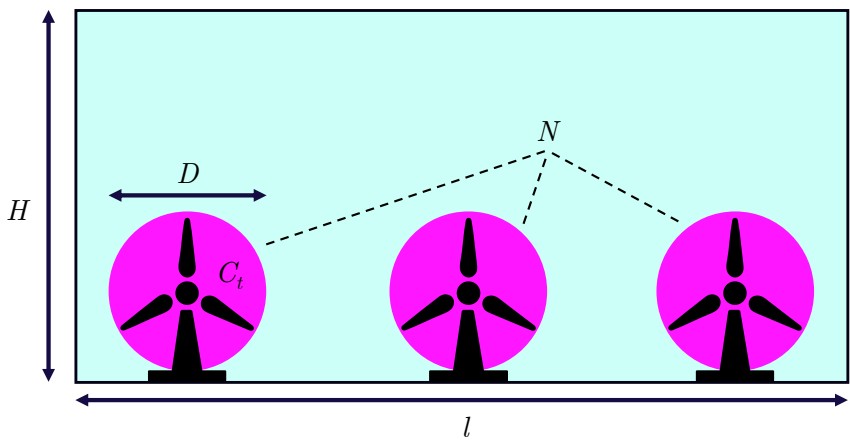

**Figure 1.** Schematic overview of the global blockage model. For an overview of symbols, see Table 1.

To derive our global blockage model, the wind farm is viewed as a large object with a given frontal - yet porous - windward facing area. The "porosity", or rather the "impermeability", of the object is determined by the swept rotor areas, inducing a blockage to the incoming winds. Based on this argument, it can be hypothesized that the relative global blockage deficit $\frac{\Delta U}{U}$ is proportional to the ratio of the effective turbine swept area ($A_{\mathrm{eff,turbines}}$) and the upstream wind inflow area ($A_{\mathrm{inflow}}$). Upon this assumption, we arrive at the following proportionality:

$$\frac{\Delta U}{U} \sim \frac{A_{\mathrm{eff,turbines}}}{A_{\mathrm{inflow}}}, \tag{1}$$

in which the right-hand side can be interpreted as the fraction of the perpendicular inflow area ($A_{\mathrm{inflow}}$) blocked by turbines ($A_{\mathrm{eff,turbines}}$). Consider a downstream wind direction such that the front row of turbines is precisely perpendicular to the wind direction. In line with to gravity-wave models such as of Smith (2010), the area below the atmospheric boundary layer will be considered. However, in our model, a fixed atmospheric boundary layer height is assumed for simplicity. Using this fixed boundary layer height, the inflow surface $A_{\mathrm{inflow}}$ can be defined as the perpendicular area spanned by a length-scale $l$ and the atmospheric boundary layer height $H$, as depicted in Figure 1. Here, $l$ is assumed to scale with the size of the wind farm. In the sequel, we will approximate $l$ as the distance between the outermost turbines in the front row perpendicular to the incoming wind. Hence, one arrives at equation (2) for the inflow wind area:

$$A_{\mathrm{inflow}} = H\, l. \tag{2}$$

The total area of front-row turbines can then be obtained by $\frac{\pi}{4} D^2 N$, where $D$ represents the rotor diameter and $N$ the number of front row turbines. Finally, to obtain the *effective* turbine swept area, we make the assumption that the effective turbine interaction on the upstream wind is proportional to the turbine thrust coefficient $C_t$, as this provides a wind speed dependent





measure of the relative momentum deficit induced by turbines. In our model formulation, this induces a relatively lower blockage effect for high wind speeds. Taking $C_t$ as a function of the inflow wind speed $U$, the following equation is obtained:

$$A_{\text{eff,turbines}} = \frac{\pi}{4} D^2 N C_t(U).$$ (3)

Combining equation (1) - (3), we propose the following expression (4) for the wind deficit of the inflow wind speed for a given wind direction:

$$\frac{\Delta U}{U} \sim \frac{\pi D^2 N C_t(U)}{4 H l}.$$ (4)

To close this proportionality relationship, we expect the blockage effect to be dependent on asset shape characteristics. In order to account for this asset shape dependency, we introduce the parameter $C_d$:

$$\frac{\Delta U}{U} = \frac{\pi C_d D^2 N C_t(U)}{4 H l}.$$ (5)

Where $C_d$ can be interpreted as a wind asset drag coefficient, depending upon the shape of the asset perpendicular to the wind direction. Higher values of $C_d$ will result in an increased global blockage. In the scope of this study, $C_d$ is assumed to be independent of wind speed, although such a dependency can be implemented if desired. Table 1 presents an overview of the variables with their associated units.

| Symbol | Unit | Description |
|--------|------|-------------|
| $U$ | $\mathrm{m\,s^{-1}}$ | Front-row inflow wind speed |
| $\Delta U$ | $\mathrm{m\,s^{-1}}$ | Global blockage front-row inflow wind speed deficit |
| $C_d$ | $-$ | Normalized asset-shape drag coefficient |
| $D$ | m | Turbine rotor diameter |
| $N$ | $-$ | Number of front row turbines facing specified wind direction |
| $C_t$ | $-$ | Turbine thrust coefficient |
| $H$ | m | Atmospheric boundary layer height |
| $l$ | m | Associated length-scale of layout facing wind direction |

**Table 1.** Overview of quantities used in the blockage model.

## 3   Wake modeling setup

In this section, the chosen wake model design choices are discussed, describing the setup for each submodel of the engineering wake model approach as used and implemented in subsequent sections.





### 3.1 Wake deficit model

Similar to the implementation of Stipa et al. (2024), the well-known Bastankhah and Porté-Agel (2014) wake deficit model is used, formulated as:

$$\frac{\Delta U}{U} = \left(1 - \sqrt{1 - \frac{C_t(U)}{8(k\,x/D + \varepsilon)^2}}\right) \exp\left(-\frac{1}{2(k\,x/D + \varepsilon)^2}\left[\left(\frac{z - z_h}{D}\right)^2 + \left(\frac{y}{D}\right)^2\right]\right), \tag{6}$$

in which $\frac{\Delta U}{U}$ represents the relative wind speed deficit, $C_t$ is the turbine's thrust coefficient, $x$, $y$ and $z$ represent downstream distance and spanwise coordinates respectively, $D$ represents the turbine diameter size, $z_h$ the hub height. $\varepsilon$ is based on a parametrized function $\beta$ of the thrust coefficient, yielding $\varepsilon = 0.2\sqrt{\beta}$. This leaves one free parameter $k$, which accounts for the wake dispersion speed and angle. In the original paper Bastankhah and Porté-Agel, $k$ is taken as $0.05$ for offshore or offshore-like cases and $k = 0.075$ for other cases. However, the commonly used wake model implementation PyWake applies a default setting of $k = 0.0324555$ (Pedersen et al., 2023).

### 3.2 Wake superposition model

To dictate how overlapping wake deficits should be combined, two main approaches include the *linear sum* and the *quadratic sum* of wakes. In the first model, wake deficits are compounded linearly, whereas the quadratic wake deficit model combines wake deficits by taking the square root of the squared overlapping deficits. In the implementation of this paper, a linear summation model is used. The models of Allaerts and Meyers (2019) and Stipa et al. (2024) also utilize a linear summation strategy, with the addition of a spatially dependent weighting of wake deficits. Furthermore, linear superposition showed to be the most consistent compared to LES data in our benchmark experimentation setups. For instance, as shown in Appendix A, using the squared sum wake superposition model gives an unrealistically small wake shape parameter $k$ in the order of $10^{-3}$, possibly due to compensation for the underestimation of wake deficit superposition effects.

### 3.3 Local blockage model

To obtain a benchmark for a widely used engineering blockage model, the numerical self-similar blockage model as proposed by Troldborg and Meyer Forsting (2017) is used, implemented in PyWake under the name *SelfSimilarity* blockage model. At the core of the model lies the assumption, supported by RANS-data, that under the correct scaling, blockage deficits are approximately similar under all chosen simulation conditions. This is achieved by taking a characteristic half-width $r_{1/2}(x)$ for which the relative blockage induction percentage $a(r,x) = 1 - U(r,x)/U_\infty$ satisfies $a(r_{1/2}(x),x) = \frac{1}{2}a(0,x)$, at which point the inductance is half of the *centerline induction* (Troldborg and Meyer Forsting, 2017). This observation is used to fit dimensionless parametrized functions to obtained RANS-simulation data. Troldborg and Meyer Forsting state that the chosen approach might not hold up in real world conditions, but does provide an elegant model, especially for distances exceeding 1 $R$ rotor distance upstream of turbines. Furthermore, no additional parameter input choices are required. A graphical overview of the wake model with the self-similar blockage is shown in Figure 2.

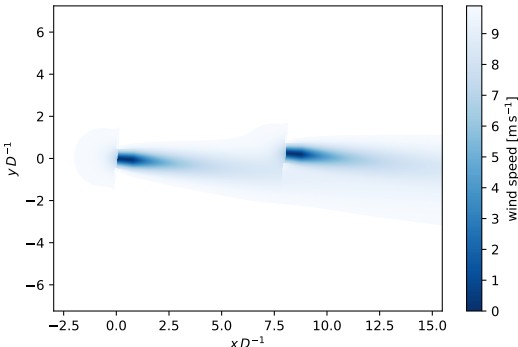

**Figure 2.** A visualization of the wake model, showing effective wind speeds at hub height for two V80 turbines ($D = 80\,\text{m}$) using the Bastankhah Gaussian wake deficit model with linear superposition and the self-similar local blockage model for a wind speed of $10\,\text{m s}^{-1}$ at $275°$.

### 3.4 Global blockage model

Some of the quantities in the global blockage model defined by Equation (5) are readily available in wake modeling implementation, while other quantities are obtained from an analysis of site or turbine conditions. The turbine diameter $D$ and the thrust coefficient $C_t$ are known and also required for the wake deficit model formulation (Equation (6)). Implementation choices have to be made for the site layout parameters $N$ and $l$. As a first approach, we propose to compute the number of front row turbines $N$ as the number of wind turbines that are not in a $30°$ conical wake shape behind another turbine. Note that $N$ as such will be a wind direction dependent quantity. This analysis is then also extended to derive the length scale $l$, which for now is taken to be the maximum shortest distance between two of the front-row turbines. It is noted that the specific expressions for $N$ and $l$ might be subject for further improvement. Finally, the atmospheric boundary layer height $H$ and shape parameter $C_d$ pose the largest modeling challenge. The boundary layer height is expected to vary with the atmospheric stability over time. As a first approach for steady-state wake models, an averaged wind direction and/or speed binned boundary layer height is chosen. The drag coefficient $C_d$ is a free parameter and can be chosen so that a perpendicular straight-lined "block" of wind turbines is expected to approximately yield $C_d = 1$ as the layout resembles that of Figure 1. If no specific data is available for $H$ or $C_d$, but energy production data is available, one strategy can be to optimize over these values, obtaining the best model fit for the respective parameters by parameter calibration. Using an implementation with a discretized number of wind directions $i \in \{1, \ldots, N_{\text{wd}}\}$ and wind speeds $j \in \{1, \ldots, N_{\text{ws}}\}$, one arrives at the following discrete formulation of (5) indexed over all wind directions $i$ and wind speed bins $j$, respectively:

$$\frac{\Delta U_{i,j}}{U_{i,j}} = \frac{\pi\, c_{d_i}\, D^2\, N_i\, C_t(U_{i,j})}{4\, H_{i,j}\, l_i}, \quad \text{for} \quad i \in \{1, ..., N_{\text{wd}}\}, \quad j \in \{1, ..., N_{\text{ws}}\}. \tag{7}$$



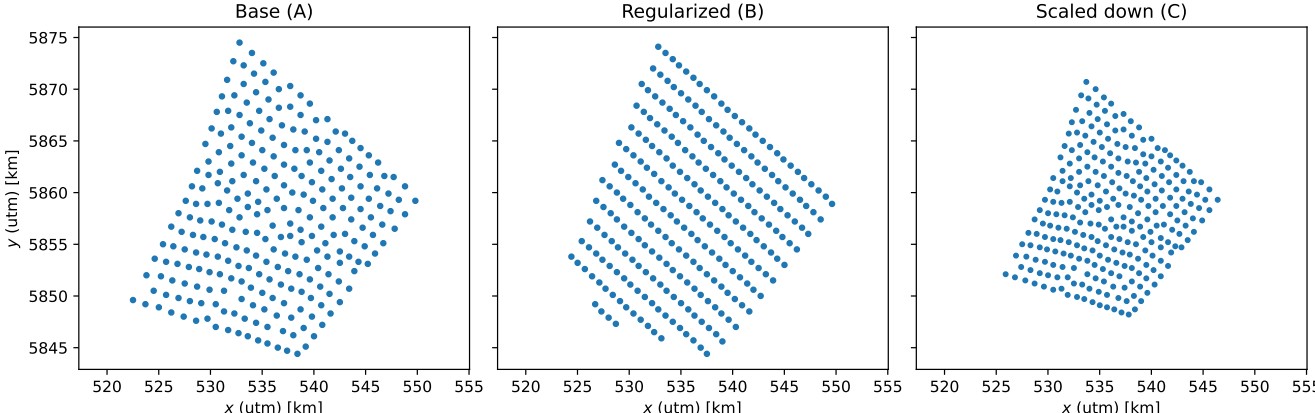

**Figure 3.** IJver base layout (A) and two variations (B), (C) within the same site boundaries as used in experimentation.

## 3.5 Solver

For all model implementations without the local self similarity blockage model as described above, the PyWake downwind propagation solver (*PropagateDownwind*) is used, calculating wind speed deficits and its superpositions in the streamwise direction. When including the self similar local model, however, an iterative solver is required. In PyWake, for instance, the solver strategy *All2AllIterative* calculates interactions between turbines and adds deficits and blockage effects iteratively until

a stable solution is obtained. Default settings are taken, yielding a stable solution requirement of a change in effective wind speed of $< 10^{-6} \, \mathrm{m \, s^{-1}}$. A major downside to this solver strategy is the large number of iterations required, especially for larger wind farms, resulting in higher computational costs, as will be demonstrated in Section 5.

## 4 Case study: IJmuiden Ver

In this section, the experimentation setup for benchmarking the parametrized global blockage model and other blockage models

as described in Section 3 is discussed. First, the wind asset site data is discussed, for which atmospheric Whiffle LES data is obtained to be used in the wake models. For the three proposed turbine layout setups, benchmark high-fidelity Whiffle LES energy production data is obtained. Finally, the wake model setup is discussed, outlining the experimentation setup with varying (global) blockage models.

### 4.1 Site data

To assess the presented global blockage model, the reference wind asset *IJmuiden Ver* will be considered as a case study, modeling a hypothetical setup for *Kavel* I - IV (Noordzeeloket, n.d.). The total installed capacity in this wind farm will be approximately 4GW.





Our hypothetical setup for this asset consists of 268 turbines of the IEA Wind 15-Megawatt Offshore Reference Wind Turbine (Gaertner et al., 2020), configured in three different layout scenarios. For the base layout (A), a uniform layout (Baas et al., 2023). Layout (B) is an adaptation of (A), using the same area constraints in a regularized setup perpendicular to northeasterly winds. This layout is taken to assess how well the blockage models carry over to more regularly perpendicular layouts. Finally, layout (C) is an adaptation of (A) with a scaled down setup, resulting in 25% smaller inter-turbine distances. This layout is proposed to study the effect of a more dense layout on the blockage models. The three layouts are depicted in Figure 3. All three layouts respect the designated area constraints for this asset.

## 4.2 LES atmospheric results

The site's atmospheric conditions are calculated using one run of Whiffle's LES model in a validated operational setting. The LES model has a domain of $70.4 \times 76.8$ km with a resolution of 100 m, nested within a meso-scale model that covers a domain of $256 \times 256$ km at 2 km resolution. In the vertical direction, the LES model has a near-surface vertical resolution of 25 m and the domain extends up to 3 km. The meso-scale simulation has a near-surface vertical resolution of 40 m resolution and covers a domain up to 8 km in height. The meso simulation is driven by initial and boundary conditions from ECMWF ERA5 reanalysis data (Hersbach et al., 2020). The resulting average wind rose and spatial wind speeds at hub-height (150m) are shown in Figure 4 and 5. Mean site wind speeds weighted averages for all wind directions are $10.97 \mathrm{~ms}^{-1}$, with a dominant wind direction from the southwest-west, reaching wind speeds of over $25 \mathrm{~ms}^{-1}$. The next most common wind direction is from the northeast, albeit with lower wind speeds. Finally, the atmospheric boundary layer height is taken as a model output

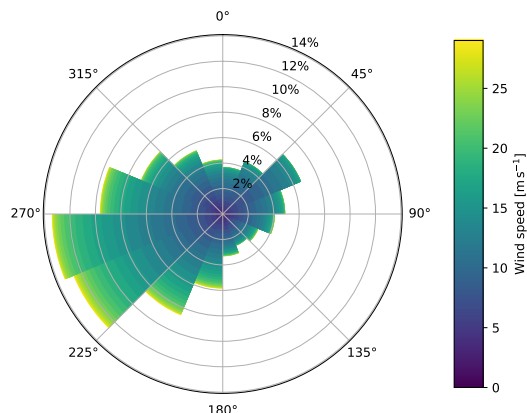

**Figure 4.** IJver site wind rose LES data, no turbine effects.

of the Whiffle LES model without turbine effects. Results grouped by wind direction are presented in Figure 6, showing values in the range of 200 - 900 m.

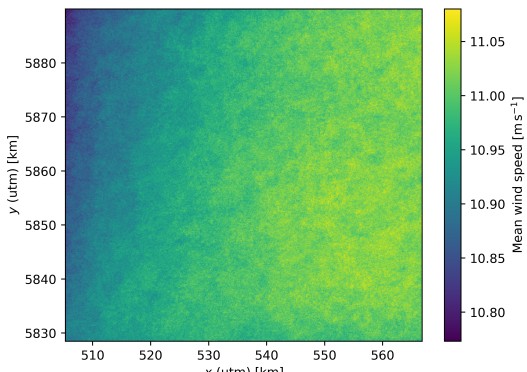

**Figure 5.** IJver site average wind speed spatial field LES data, no turbine effects.

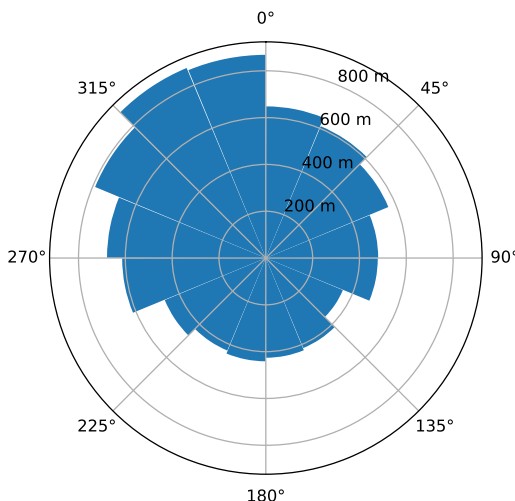

**Figure 6.** IJver site LES average atmospheric boundary layer height per wind direction, no turbine effects.

### 4.3 LES energy production results

LES energy production results with turbine effects are obtained using the same (ERA5) input data and LES model setup for all site layouts as described in Section 4.1. Hence, three LES energy production model runs are carried out. Wind turbines are modeled using the actuator disk model, of which a sensitivity analysis using this site setup in LES is given by Baas et al. (2023). These results will serve as the benchmark and training data for the engineering wake models with different blockage models. In Figure 7, average yearly wind velocity deficits generated by turbine interactions are presented for base layout (A). A clear blockage effect can be observed, with different behavior dependent on the frontal shape of the asset as proposed in the formulation of the global blockage model. This provides further motivation for introducing a shape-dependent drag coefficient





**Figure 7.** LES average wind speed deficits per 30-degree wind direction sectors for the base layout (A). Deficits are computed as the relative difference between the LES wind speed results without turbine interactions and LES results including turbine effects.

parameter $C_d$. Furthermore, similar to the findings of Maheshwari et al. (2024), a speedup effect around the wind farm can be observed for a number of wind conditions.

|   | Layout | Wind directions | $k$ calibration |
|---|---|---|---|
| A | Base layout | Northeast | Training set |
| B | Regularized layout | Northeast | Test set |
| C | Downscaled layout | Northeast | Test set |
| A* | Base layout | All | Test set |

**Table 2.** Overview of experimentation site data setup. All setups are ran without any blockage (0), with self-similar local blockage model (1) and with the parametrized global blockage model (2), with respective fits of $k$ obtained in setup A and tested on all other setups.





## 4.4 Wake model experimentation setup

To apply and compare the global blockage model, one wake parameter calibration setup and three blockage model test experiments are conducted. In all setups, the energy production results of the engineering wake model setup as outlined in Section 3 are compared to LES turbine energy production results. As input site wind data for the wake model energy production calculations, LES atmospheric data without turbine effects is used as described in Section 4.2, implemented using Wind Resource Grid (WRG) spatially Weibull-distributed wind roses, interpolated to a $400 \times 400$ m grid. For all experiments, turbulence intensity (TI) is taken as the average obtained LES value of $3.5\%$ at hub-height.

An overview of experimentation setups is presented in Table 2. One parameter calibration training setup (A) is run initially, taking the base layout restricted to northeastern wind directions $(0 - 90°)$. For this training case, the Bastankhah and Porté-Agel (2014) wake deficit model parameter $k$ is fitted to Whiffle LES energy production data for the same northeastern wind directions. The parameter calibration is conducted using a gradient-descent algorithm minimizing the bias in energy production output of the wake model implementation compared to the LES energy production data with turbine effects. The northeastern wind is taken since the site layout is relatively perpendicular to this wind direction and has a straight front-row shape structure, satisfying the assumptions for the global blockage model, where $C_d = 1$ is taken for simplicity. This is further motivated by the regular appearance of blockage behavior in the LES data, as seen in Figure 7. The northeastern wind speeds make up $20.39\%$ of the yearly weather and have an average atmospheric boundary layer height $H$ of $575$ m.

The obtained values for $k$ are then applied to the regularized layout (B), which is conjectured to possibly pose a more stringent blockage effect by the row-wise layout but has the approximate same structure and calculations for the parametrized global blockage model compared to base layout (A). Subsequently, the same setup is tested on scaled-down layout (C), testing the consistency of the blockage models when taking a more dense configuration. Finally, the model is applied to layout A using all wind directions in experimentation A*. In this last case, the average atmospheric boundary layer $H$ is varied and obtained from LES-data for wind direction sections of $90°$, yielding values of $575, 425, 492$ and $781$ m for northeasterly, southeasterly, southwesterly and northwesterly winds respectively for the global blockage model. Here, a shape parameter $C_d = 1$ is taken for all wind directions as in the other layouts. The global blockage formulation (7) is then used and implemented in PyWake through the `Speedup` key in the dataset of the `Site` object.

Experiments are run without blockage model (0), with SelfSimilarity (1) and the global blockage model (2), yielding different parameter values of $k$. In testing scenarios B, C and A*, wake model output energy production data is compared to LES values for error statistics. For experimentations A, B and C, only one northeasterly wind direction bin is used. In these setups, we use the following error metrics:

$$\text{RMSE} = \sqrt{\frac{\sum_{i=1}^{N}(y_i - x_i)^2}{N}} \quad \text{and} \quad R^2 = 1 - \frac{\sum_{i=1}^{N}(y_i - x_i)^2}{\sum_{i=1}^{N}(\bar{\mathbf{x}} - x_i)^2}, \quad \text{where } \bar{\mathbf{x}} = \frac{\sum_{i}^{N} x_i}{N}, \tag{8}$$

In which $\mathbf{x}$ are the LES energy production observations and $\mathbf{y}$ indicate the wake model predictions for the $N$ individual turbines.

For the experiments carried out as specified for the multi-bin wind setups in A*, two choices in error metric are taken for wind direction sectors $j \in \{0, \dots, M\}$. The aggregated (agg) error metrics show results when the directional summation is





carried out first:

$$\text{RMSE}_{\text{agg}} = \sqrt{\frac{\sum_{i=1}^{N}(\sum_{j=0}^{M}[y_{i,j} - x_{i,j}])^2}{N}} \quad \text{and} \quad R_{\text{agg}}^2 := 1 - \frac{\sum_{i=1}^{N}(\sum_{j=0}^{M}[y_{i,j} - x_{i,j}])^2}{\sum_{i=1}^{N}(\sum_{j=0}^{M}[\bar{\mathbf{x}}_j - x_{i,j}])^2}, \quad \text{where} \quad \bar{\mathbf{x}}_j = \frac{\sum_{i=1}^{N} x_{i,j}}{N}. \quad (9)$$

Conversely, for wind-direction specific error metrics (wd), the directional summation is carried out last, i.e.:

$$\text{RMSE}_{\text{wd}} = \sum_{j=0}^{M} \sqrt{\frac{\sum_{i=1}^{N}(y_{i,j} - x_{i,j})^2}{N}} \quad \text{and} \quad R_{\text{wd}}^2 := 1 - \frac{\sum_{j=0}^{M}\sum_{i=0}^{N}(y_{i,j} - x_{i,j})^2}{\sum_{j=0}^{M}\sum_{i=0}^{N}(\bar{\mathbf{x}}_j - x_{i,j})^2}, \quad \text{where} \quad \bar{\mathbf{x}}_j = \frac{\sum_{i=1}^{N} x_{i,j}}{N}. \quad (10)$$

## 5   Results

| | Blockage model | Fitted $k$-value | Energy Production (GWh) | Bias | RMSE | $R^2$ | Time (s) |
|---|---|---|---|---|---|---|---|
| (A0) | - | 0.0131 | 2464.34 | -0.01 % | 9.05% | 0.730 | 0.67 |
| (A1) | SelfSimilarity | 0.0132 | 2464.96 | +0.02% | 8.46% | 0.764 | 73.84 |
| (A2) | GlobalBlockage | 0.0210 | 2464.54 | +0.00% | 4.56% | 0.931 | 0.66 |

**Table 3.** Results of wake deficit parameter training setup (base layout A, northeast). Computational time is shown for one model run. Note that the RMSE of this setup can be interpreted as the *residual error* in training.

In Table 3, results of the northeast wind parameter calibration training setup are presented. Error metrics root mean squared error (RMSE) and $R^2$ are defined as in (8). The spatial distribution of errors is vizualized in Figure 9 and scatter plots are provided in Figure 11. All models are able to converge to the total asset-wide energy production as obtained by the LES model, satisfying the bias within a tolerance of 1 GWh. By comparison, using the default value of $k = 0.05$ as proposed by Bastankhah and Porté-Agel (2014) yields a very different energy production output, as shown in Appendix B. Clearly, a large difference in calibrated parameter $k$ value can be observed, with a value for the global blockage model (A2), nearly double that of the local self-similar (A1) and no-blockage case (A0). Although much closer, this value of $k = 0.0210$ is still lower compared to default literature values as described in Section 3.

A spatial interpretation of parameter training residuals compared to LES data is provided in Figure 8. Whereas both the self-similar local blockage model and parametrized global blockage model overpredict in the front row on average, the effect is much less when applying the global blockage model. In the "far wake" further downstream, both models yield an underprediction. However, when applying the global blockage model, this underprediction is considerably smaller, which most likely can be attributed to the higher value of $k$, creating a less stringent wake deficit effect. On the whole, the self-similar local blockage model obtains a slight error improvement compared to no blockage model, with an $R^2$ of 0.764 versus 0.730. The parametrized global blockage model however is much more accurate, with an $R^2$ of 0.931 and an RMSE of only 4.56%. Computational times for (A0) - no blockage - and (A2) - parametrized global blockage model- are similar. However, the self-similar local blockage model of (A1) requires much more computational time, by a factor of about 100 compared to (A0) and (A2). This is due to the requirement of the iterative solver as discussed in Section 3.





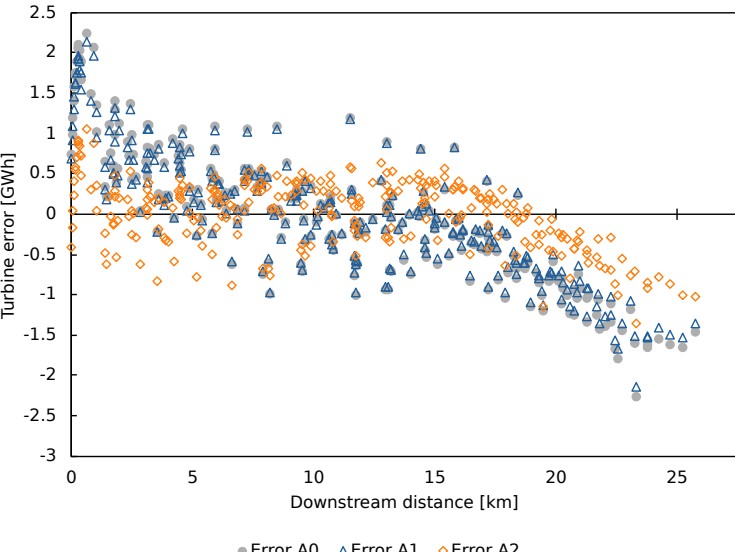

**Figure 8.** Errors per wind turbine in average streamwise direction ($45°$), using no blockage model (A0), using the self-similar local blockage model (A1) and using the parametrized global blockage model (A2).

|  | Blockage model | Energy Production (GWh) | Bias | RMSE | $R^2$ | Time (s) |
|---|---|---|---|---|---|---|
| (B0) | - | 2470.53 | +0.3% | 10.69% | 0.622 | 0.55 |
| (B1) | SelfSimilarity | 2470.45 | +0.3% | 10.07% | 0.664 | 39.57 |
| (B2) | GlobalBlockage | 2410.87 | -2.1% | 5.52% | 0.904 | 0.60 |

**Table 4.** Results of test experiment B (regularized layout, northeast).

Applying the parameter values of $k$ found in training setup A to experiments B using the regularized layout gives results as presented in Table 4. In general, errors are higher compared to the residuals of test setup (A). The energy production is nearly
250 exactly matched by the blockage-less and self-similar local blockage model, with an underestimation of 2% when using the parametrized global blockage model. Hence, the bias is higher when using the parametrized global blockage model, which might indicate that the blockage effect is too strong compared to the found wake shape parameter $k$. Prediction errors however are much lower when applying the parametrized global blockage model, yielding an improvement in wake model prediction errors as shown spatially in Figure 9.
The same experimentation is carried out on the downscaled, more dense setup (C), of which results are presented in Table 5. A notably higher error is obtained compared to the residuals in training set A. All models underpredict the energy production with 2 - 3 %, with the parametrized global blockage model in (C2) resulting in the most severe underprediction. Errors however are much lower when using the global blockage model, which is still able to obtain an $R^2$ of 0.828, compared to an $R^2$ of 0.559 for (C1), using the self-similarity approach. As can be seen in Figure 9, overprediction in the front rows in scaled down layout





|  | Blockage model | Energy Production (GWh) | Bias | RMSE | $R^2$ | Time (s) |
|---|---|---|---|---|---|---|
| (C0) | - | 1957.04 | -2.5% | 18.19% | 0.501 | 0.54 |
| (C1) | SelfSimilarity | 1951.39 | -2.8% | 17.14% | 0.559 | 42.53 |
| (C2) | GlobalBlockage | 1946.79 | -3.0% | 10.72% | 0.828 | 0.55 |

**Table 5.** Results of test experiment C (downscaled layout, northeast).

(C) is relatively higher compared to base layout (A), which might indicate that the global blockage effect is underestimated for a more dense configuration. Another possibility is that the balance between the blockage and wake effect does not scale appropriately compared to the calibration setup (A), possibly due to the choice of length scale $l$. However, as seen by the lower errors of (C2) compared to (C0) and (C1), this scaling occurs much more accurately when applying the parametrized global blockage model. Computational times are similar to training case (A).

|  | Blockage model | Energy Production (GWh) | Bias | $\text{RMSE}_{\text{agg}}$ | $R^2_{\text{agg}}$ | $\text{RMSE}_{\text{wd}}$ | $R^2_{\text{wd}}$ | Time (s) |
|---|---|---|---|---|---|---|---|---|
| (A0*) | - | 16675.44 | +0.4% | 1.66% | 0.940 | 3.80% | 0.573 | 8.30 |
| (A1*) | SelfSimilarity | 16677.32 | +0.4% | 1.61% | 0.943 | 3.65 % | 0.607 | 477.00 |
| (A2*) | GlobalBlockage | 16784.52 | +1.1% | 1.78% | 0.930 | 2.70 % | 0.784 | 9.30 |

**Table 6.** Results of test experiments A* (base layout, all wind directions).

Finally, the obtained parameter values for $k$ of training set (A) is applied to all wind directions on the same layout for (A*). Results are presented in Table 6. The total energy production prediction results appear satisfactory with an overprediction bias of the order of $0$ - $1$ %. Interestingly however, contrary to the results of (B2), the global blockage model in this site setup (A2*) creates an overprediction bias instead of an underprediction, most likely due to the different wind conditions or perpendicular asset shape properties.

In terms of prediction error however, a more subtle pattern emerges. At first sight, it appears all blockage models perform very well. When energy production results are first aggregated as the sum over all four wind direction sectors, root mean squared errors $\text{RMSE}_{\text{agg}}$ are in the range of $1$ - $2$ % and an $R^2_{\text{agg}}$ of $0.93$ - $0.94$, with calculations as in Equation (9). The parametrized global blockage model, using a fixed default value of $C_d = 1$, performs slightly worse compared to the other models. Spatial results in Figure 9 show a clear overprediction in the dominant wind direction for (A0*) and (A1*), with a very different pattern using the parametrized global blockage model for (A2*).

However, when calculating errors per wind direction first before summation, a different effect is observed. We denote these error metrics by $\text{RMSE}_{\text{wd}}$ and $R^2_{\text{wd}}$ as outlined in equation (10). Using these metrics, it can be seen that errors cancel out more strongly in the aggregated case for (A0*) and (A1*) compared to using the parametrized global blockage model in (A2*), where more accurate results are obtained per wind direction sector. This effect is visualized by the sum of the absolute error per wind direction, shown in Figure 10.





An important consequence of this result is in the interpretation to model validity for specific setups. Whereas the global blockage model does not appear to add much improvement in accuracy for total aggregated production values, a big error reduction can be observed when assessing the error over the specific sectors of wind directions. This essentially shows model validity is improved for specific wind condition inputs, posing a motivation for the wind-direction dependent implementation

of the parametrized global blockage model and the effect of the model in general.



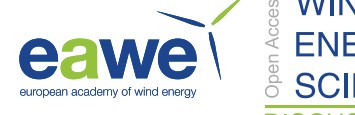

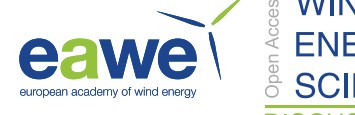

**Figure 9.** Spatial distribution of LES and wake model energy production (EP) prediction errors per turbine for all site layouts, using no blockage model (0), the self-similar local blockage model (1) and the parametrized global blockage model (2). Note that for A*, the error can be interpreted as the aggregated approach as taken in RMSE$_{agg}$ of (9) per turbine $i \in \{1, \ldots, N\}$.




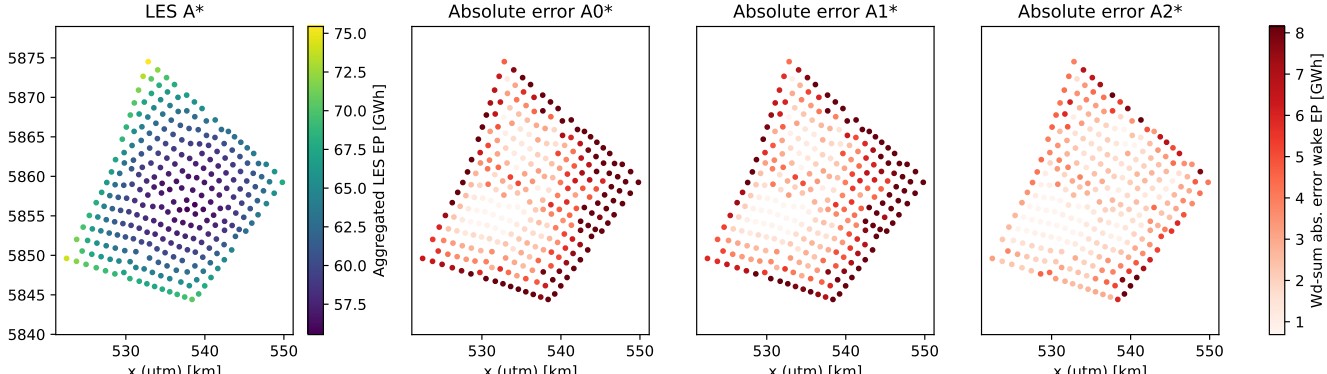

**Figure 10.** Spatial distribution of LES and wake model energy production (EP) sum of absolute errors over all 4 wind direction sectors. This error can be interpreted as the same approach as taken in RMSE$_{wd}$ of (10) per turbine $i \in \{1, \ldots, N\}$. Wake model results are shown using no blockage model (A0*), the self-similar local blockage model (A1*) and the parametrized global blockage model (A2*).

## 6 Conclusions

This paper proposes a new wake-model integrated parametrized model for global blockage. The model is formulated and tested in various example offshore site layouts, benchmarked against high-fidelity LES data with a calibrated wake deficit parameter.

Results show that the parametrized global blockage model improves accuracy compared to both engineering wake models
without blockage and models with turbine-level self-similar local blockage model implementation. Furthermore, the blockage model can be run with much lower computational cost compared to local blockage effects, for which a more computationally demanding iterative solver is required.

Another main advantage of the parametrized global blockage model is the simplicity in parameter space. Provided that data or an estimate of the atmospheric boundary layer height $H$ is available, only the shape-dependent drag parameter $C_d$ and length
scale $l$ have to be evaluated and possibly calibrated for site conditions. For a simple and regularly shaped asset, $C_d = 1$ appears to yield satisfactory results in the experimentation setup of this paper. We conjecture that even in the absence of high-quality estimates for these input parameters, the parametrized global blockage model is still fit for use in improving wake model energy production model fidelity.

One major indication pointing towards this improvement is the profound effect on the wake deficit shape parameter $k$. In
all results without the parametrized global blockage model, a much lower value of the wake shape parameter $k$ is obtained after calibration to LES data, indicating a more narrow and deficit-inducing wake shape. This is most likely due to the fact that without blockage effect, the engineering wake model is "overcompensating" the wake deficit to obtain a calibrated energy production output since the (global) blockage deficit is underestimated. Hence, including an accurate blockage effect is expected to also improve wake deficit model validity. We expect this can be further improved, as current results still show an
underestimation in the far wake, as can for instance be seen in Figure 8.



**Figure 11.** Scatter plot of turbine energy production of wake model vs. LES, using no blockage model (0), the self-similar local blockage model (1) and the parametrized global blockage model (2). As discussed in Section 5, values for A* are obtained using aggregated data over all wind directions and thus appear more correlated compared to the correlation per wind direction.





Finally, another indication to wake model validity improvement can be seen when testing the model in multiple wind direction sectors. When the (global) blockage effect is underestimated, the spatial error distribution, overestimating in the front rows and underestimating in the far wake, can lead to error cancellation when taking aggregated results over multiple wind directions. This can lead to a deceptively high appearance of correlation at turbine level, which is shown in our test results to

not hold when assessed for wind directions separately.

Several recommendations for future work can be made to improve the introduced model and further study the behavior of global blockage effects:

The current parametrization poses unknowns in terms of the parameter space and dependencies. More research is required to assess the impact of $C_d$ selection and calibration. For $l$, a choice of a function to obtain to obtain values suitable for general

asset shapes and scales can be developed and tested on more layouts. A possibility in the current parameter setup would be to conduct a large scale parameter optimization experiment, possibly extending the dependency of parameters to wind speed and / or stability variables as well.

Furthermore, currently the test setups are carried out on offshore sites with a simple asset structure. The influence of terrain interaction could pose a problem for the current parametrization setup. The same holds for asset shape requirements, as strongly

nonconvex shapes may be unsuitable for the current formulation.

Finally, as shown by LES results and existing research, speedup effects can be observed at the edges of assets. More research can be conducted to model this phenomenon in an efficient, possibly parametrized approach.

## Appendix A: Squared sum wake superposition

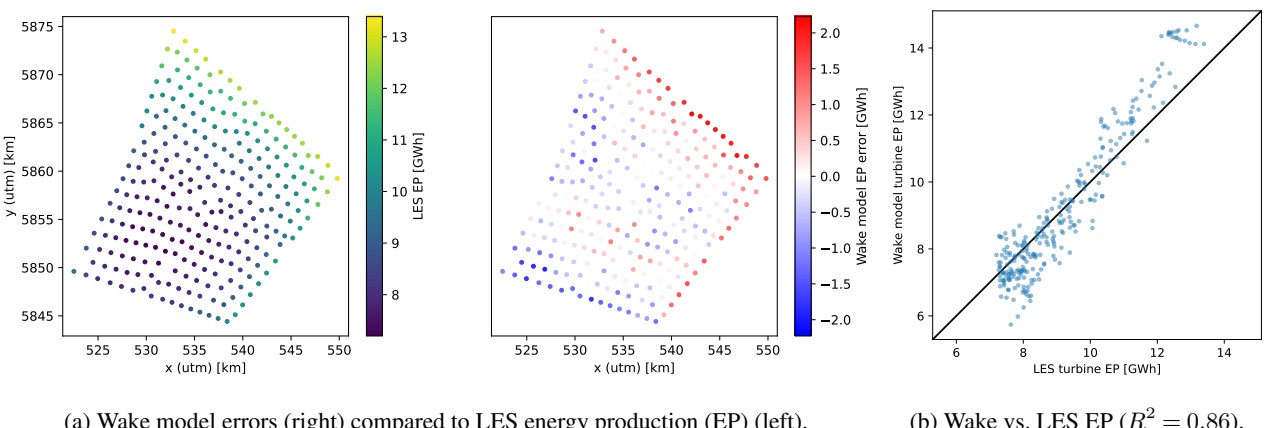

(a) Wake model errors (right) compared to LES energy production (EP) (left).    (b) Wake vs. LES EP ($R^2 = 0.86$).

**Figure A1.** Bias-calibrated engineering wake model (energy production $= 2465.22$ GWh, $k = 0.0034$) energy production prediction errors using squared sum wake superposition. All other design settings are taken as in setup (A0) of Section 4.4

.

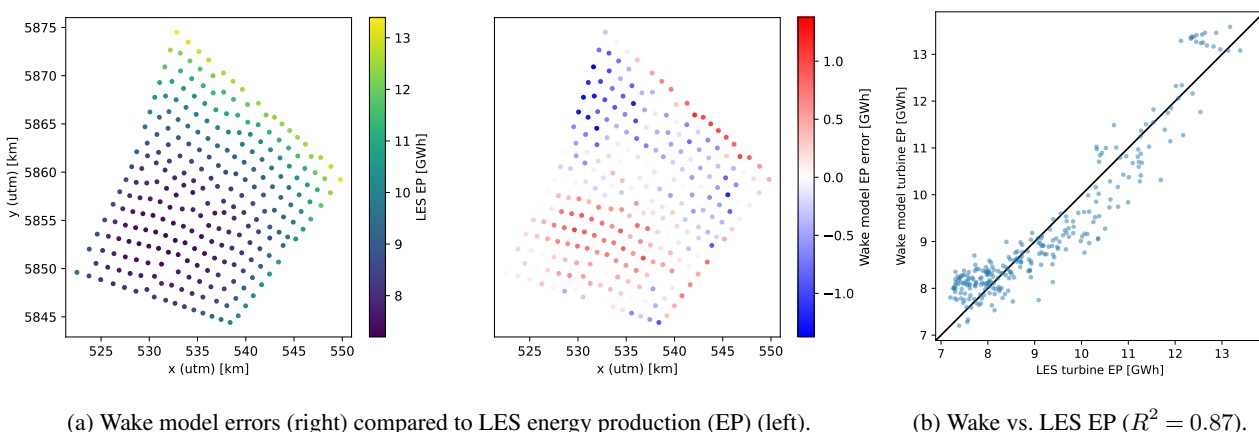

(a) Wake model errors (right) compared to LES energy production (EP) (left).     (b) Wake vs. LES EP ($R^2 = 0.87$).

**Figure A2.** Bias-calibrated engineering wake model with parametrized global blockage (energy production $= 2464.26$ GWh, $k = 0.0090$) energy production prediction errors using squared sum wake superposition. All other design settings are taken as in setup (A2) of Section 4.4

.

## Appendix B: Default wake shape parameter $k$

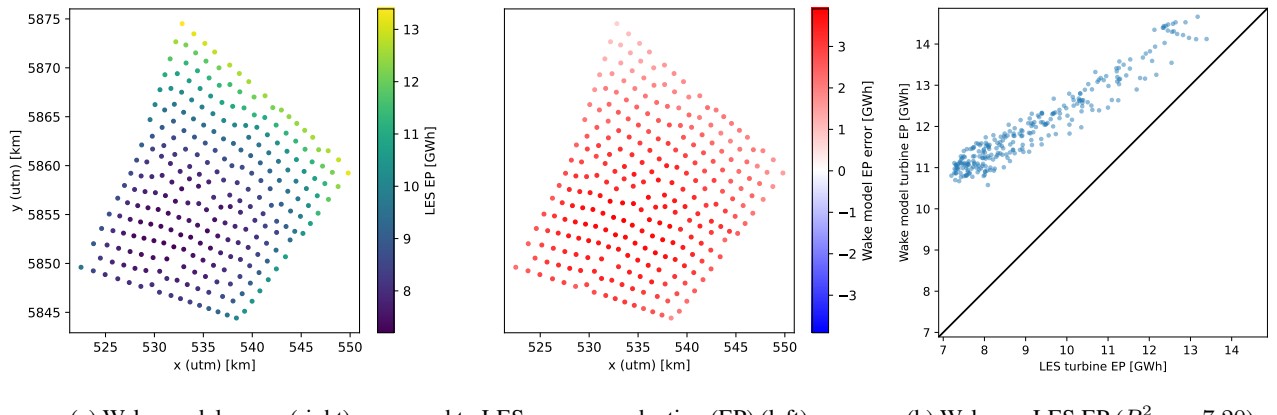

(a) Wake model errors (right) compared to LES energy production (EP) (left).     (b) Wake vs. LES EP ($R^2 = -7.20$).

**Figure B1.** Engineering wake model energy production prediction errors using wake deficit expansion parameter $k = 0.05$ as presented by Bastankhah and Porté-Agel (2014). All other design settings are taken as in (A0) of Section 4.4

.

*Author contributions.* NG: wake modeling implementation and preparation of the draft, MF: simulations and plots LES, JM: supervision and review and editing of draft. NG, MF and JM have all read and agreed to the final version of the manuscript.

*Competing interests.* All authors are employed at Whiffle Precision Weather Forecasting BV, a limited liability company.



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
