# Peer review of "A global blockage parametrization for engineering wake models"

_Wind Energy Science, 2025_

## Referee Comment (RC1)

Review comments on "A global blockage parametrization for engineering wake models" (wes-2025-71)

This paper presents a very simple engineering model of wind speed reduction upstream of a large wind farm due to the so-called "global" (or wind farm-scale) blockage effect, to be used together with a traditional engineering turbine-wake model to predict the farm power. The proposed model is (loosely) based on the concept of a whole wind farm acting like a large porous medium to reduce the wind speed upstream of it. This is a different type of model from existing "local" (or wind turbine-scale) blockage models and, as the authors demonstrate in the paper, it allows a much faster prediction of wind speed reduction upstream of a large wind farm compared to existing local blockage models.

However, the idea of modelling the farm-scale wind speed reduction as a whole (and using it together with an engineering turbine-wake model) has already been explored extensively in the literature, e.g., the "coupled wake boundary layer" model (Stevens et al. 2015, 2016) and the "two-scale momentum" model (Nishino and Dunstan 2020, Kirby et al. 2023, Legris et al. 2023). Compared to these physics-based models derived from the conservation of momentum, the global blockage model proposed here is not based on the principles of fluid mechanics and relies heavily on empirical tuning (or fitting) of the model.

A fundamental issue in the proposed approach is that the wake expansion rate "k" in the turbine-wake model needs to be tuned for a given wind farm, using its power production data (which may not be available when we need to predict the power of a future wind farm). As shown in the paper (see Table 3), the value of k required to achieve the correct farm power is significantly different between the case without using the global blockage model (k = 0.0131) and the case using the global blockage model (k = 0.0210). This means that, although k is a parameter in the turbine-wake model, this tuning process is to data-fit the global blockage model itself (as the authors also mention in their conclusions near the bottom of Page 17). The other key parameter in the proposed model, namely the farm's drag coefficient "C\_d", is not tuned in this study (C\_d = 1 is assumed in this study) and still the proposed model appears to give satisfactory results, but this does not mean that the global blockage effect is insensitive to the value of C\_d. Instead, this means that the global blockage effect, which is a farm-scale effect and should be parameterised through the farm-scale model parameters C\_d and l, has been data-fitted entirely through the tuning of the turbine-wake expansion rate k. The authors have applied their tuned k values (fitted to power production data for the original farm layout A) to different farm layouts (B, C and A\* as summarised in Table 2) to try to validate this approach, but these layouts (B, C and A\*) are similar to the original layout (A) to which the k values were fitted.

As shown by recent LES studies (e.g. Lanzilao et al. 2024), the reduction of farm-upstream wind speed depends significantly on the characteristics of the atmospheric boundary layer (as well as on the design of the wind farm itself) since the global blockage effect depends significantly on the characteristics of farm-induced gravity waves (among others). This suggests that, to sufficiently tune or train the proposed global blockage model (via the fitting of the parameter k) for different wind farms and for different atmospheric conditions, we would need the farm's power production data for those different wind farm scenarios. Since it would be difficult (or computationally expensive) to have power production data for a range of different scenarios, I suspect that the applicability of the proposed approach to different wind farm scenarios would be rather limited.

Based on the above reasons, I feel that this paper is not suitable for publication in Wind Energy Science. However, I do understand that the aim of this paper is to propose a simple and fast engineering global blockage model, which is needed in today's wind industry, and the proposed model may partially satisfy that need. I would therefore suggest a major revision of the current manuscript, taking into account the above points.

I would also suggest that the authors refine and proofread the whole manuscript before submitting it for further review. I am afraid that I found it difficult to read many parts of the current manuscript, largely due to inaccurate choice of words and unclear explanations.

Other minor comments/questions:

- How do you define the ABL height (i.e. how do you calculate "H" from your LES data) in this study? There are several different ways to define the ABL height.
- The near-surface vertical grid resolutions adopted in this study (25m in the LES and 40m in the meso-scale simulations) seem rather coarse compared to other studies. Could you justify the use of these coarse grids?
- Also, the horizontal grid resolution adopted in the LES (100m) seems rather coarse, considering that the turbines are represented using an actuator disc model. The authors refer to Baas et al. (2023) for the LES set up, but I suggest more details of the LES be included in the present paper.
- Appendices A and B do not have any text. I would suggest at least a few sentences be included in each appendix to explain about the figures presented there.

References

Kirby, A., Dunstan, T. D. and Nishino, T. (2023) An analytical model of momentum availability for predicting large wind farm power, J. Fluid Mech. 976, A24.

Lanzilao, L. and Meyers, J. (2024) A parametric large-eddy simulation study of wind-farm blockage and gravity waves in conventionally neutral boundary layers, J. Fluid Mech. 979, A54.

Legris, L., Pahus, M. L., Nishino, T. and Perez-Campos, E. (2023) Prediction and mitigation of wind farm blockage losses considering mesoscale atmospheric response, Energies 16, 386.

Nishino, T. and Dunstan, T. D. (2020) Two-scale momentum theory for time-dependent modelling of large wind farms, J. Fluid Mech. 894, A2.

Stevens, R. J. A. M., Gayme, D. F. and Meneveau, C. (2015) Coupled wake boundary layer model of wind-farms, J. Renew. Sustain. Energy 7, 023115.

Stevens, R. J. A. M., Gayme, D. F. and Meneveau, C. (2016) Generalized coupled wake boundary layer model: applications and comparisons with field and LES data for two wind farms, Wind Energy 19, 2023-2040.

---

## Author Comment (AC1)

Dear editor,

First of all, we would like to express our gratitude for the time and effort of the editor and reviewers and would like to thank the editor for considering our paper for publication in Wind Energy Science and providing us the time and opportunity to improve our manuscript.

We thank Reviewer 1 for the helpful comments and remarks on the manuscript. We appreciate the insights from the conceptual modeling point of view and would like to elaborate on how we incorporated these to improve our manuscript.

1.  **Comment:** Compared to these physics-based models derived from the conservation of momentum, the global blockage model proposed here is not based on the principles of fluid mechanics and relies heavily on empirical tuning (or fitting) of the model. A fundamental issue in the proposed approach is that the wake expansion rate "k" in the turbine-wake model needs to be tuned for a given wind farm, using its power production data (which may not be available when we need to predict the power of a future wind farm). As shown in the paper (see Table 3), the value of k required to achieve the correct farm power is significantly different between the case without using the global blockage model (k = 0.0131) and the case using the global blockage model (k = 0.0210). This means that, although k is a parameter in the turbine-wake model, this tuning process is to data-fit the global blockage model itself (as the authors also mention in their conclusions near the bottom of Page 17).
    **Response:** We agree that these parametrized wake and global models are indeed a simplification of physical phenomena. However, exactly because of this simplified structure and their ubiquitous use in industry, we are convinced of the need for explorative research in extending the parametrized approach to also include global blockage. The sensitivity of the k-parameter to the global blockage parametrization illustrates in our opinion the importance of developing a more extensive parametrized modeling approach. When taking a wake-only approach and fitting on data, parameter *k* essentially has to accommodate for this lack of global blockage effect, as we discuss in our results and conclusions.

2.  **Comment:** This means that, although k is a parameter in the turbine-wake model, this tuning process is to data-fit the global blockage model itself (as the authors also mention in their conclusions near the bottom of Page 17). The other key parameter in the proposed model, namely the farm's drag coefficient "C_d", is not tuned in this study (C_d = 1 is assumed in this study) and still the proposed model appears to give satisfactory results, but this does not mean that the global blockage effect is insensitive to the value of C_d.
    **Response:** First of all, we would like to emphasize that as of now the global blockage model and wake model take a sequential order of operations and are not coupled: the global blockage model outputs a reduction in incoming wind speed, which is then used as input for the parametrized wake model. To further highlight this, we have included a diagram highlighting the current setup in our manuscript in Section 3. We do agree that in our first version, the assumption having global blockage

parameter C_d = 1 constant leads to an experiment setup with a parameter fit on wake shape parameter k only. Even though this setup already highlights an interesting shortcoming in the classical no-blockage wake modelling setup, we agree that varying C_d increases the model validity. Therefore, we have changed our global blockage experimentation setup to also optimize on the global blockage parameter C_d. Taking the wake shape parameter as fitted in the non-global blockage case and C_d = 1 as starting point, we find similar values when fitted to our LES data, with similar order of magnitude increase in wake shape parameter k and a value of C_d in the order of magnitude of 1. This leads to a more robust conceptual result, providing a fully "data-fitted" approach for all unknown parameters in our current modeling setup, and we would like to thank Reviewer 1 for highlighting this. To showcase the sensitivity on C_d, we have included a grid-search showcasing sensitivity on C_d and k in Figure 9 of the manuscript.

As an aside, please note that this two-parameter setup requires an optimization over the mean squared error to prevent ill-posedness of the calibration step. For clarity and comparison, we have thus changed the calibration goal for all setups to minimize the mean squared error instead of the bias.

3. **Comment:** As shown by recent LES studies (e.g. Lanzilao et al. 2024), the reduction of farm-upstream wind speed depends significantly on the characteristics of the atmospheric boundary layer (as well as on the design of the wind farm itself) since the global blockage effect depends significantly on the characteristics of farm-induced gravity waves (among others). This suggests that, to sufficiently tune or train the proposed global blockage model (via the fitting of the parameter k) for different wind farms and for different atmospheric conditions, we would need the farm's power production data for those different wind farm scenarios. Since it would be difficult (or computationally expensive) to have power production data for a range of different scenarios, I suspect that the applicability of the proposed approach to different wind farm scenarios would be rather limited.

**Response:** We would like to highlight that this fundamental issue in calibrating k is manifest in the whole field of parametrized engineering wake modeling (see for instance the work of Binsbergen et al (2024) and Teng and Markfort (2020) included in our references). In practice, without operational data, this obstacle of fitting the wake shape as well as global blockage parameters can be overcome by calibrating on high-fidelity LES production and atmospheric data such as carried out in the manuscript. As we present in the results and as discussed in the conclusions section of our manuscript, we show that incorporating our parametrized global blockage approach considerably improves validity of the parametrized model when applied to different layouts and atmospheric conditions. This is demonstrated in the manuscript in the result of model test setups B, C and A*. This thus enables a smaller calibration training setup from which a model parametrization can be derived for variations in the site layout and atmospheric conditions. We agree with reviewer 1 that more extensive research is required for extending the validity of our approach to conditions differing significantly from the training setup.

4. **Comment:** I would also suggest that the authors refine and proofread the whole manuscript before submitting it for further review. I am afraid that I found it difficult to read many parts of the current manuscript, largely due to inaccurate choice of words and unclear explanations.
   **Response:** We have revised the writing in all sections and would like to thank reviewer 1 for pointing this out.

5. **Comment:** How do you define the ABL height (i.e. how do you calculate "H" from your LES data) in this study? There are several different ways to define the ABL height.
   **Response:** We have highlighted our method of estimating the boundary layer height in Section 4.2 in the manuscript for additional clarity. We would like to thank Reviewer 1 for pointing this out.

6. **Comment:** The near-surface vertical grid resolutions adopted in this study (25m in the LES and 40m in the meso-scale simulations) seem rather coarse compared to other studies. Could you justify the use of these coarse grids?
   **Response:** We have added two references to validation studies that used similar simulation settings for wind resource assessment to Section 4.2. Moreover, we reason the use of the coarse grid for the actuator disc method in Section 4.3.

7. **Comment:** Also, the horizontal grid resolution adopted in the LES (100m) seems rather coarse, considering that the turbines are represented using an actuator disc model. The authors refer to Baas et al. (2023) for the LES set up, but I suggest more details of the LES be included in the present paper.
   **Response:** Please see previous response.

8. **Comment:** Appendices A and B do not have any text. I would suggest at least a few sentences be included in each appendix to explain about the figures presented there.
   **Response:** We have added a short description and thank the reviewer for pointing this out.

---

## Author Comment (AC2)

Dear editor,

First of all, we would like to express our gratitude for the time and effort of the editor and reviewers and would like to thank the editor for considering our paper for publication in Wind Energy Science and providing us the time and opportunity to improve our manuscript.

We thank Reviewer 2 for their helpful remarks on our manuscript. Below, we will highlight our insights on the comments and several ways in which we have incorporated suggestions to improve our manuscript.

1. **Comment:** The authors make a point of the local blockage model being much slower than the wake model either alone or combined with the new global blockage model. But this hinges on using a default setting in PyWake for the convergence of the effective wind speed. Specifically, that the effective wind speed is changing by less than 1e-6 between successive iterations. This is a ridiculously small change given the accuracy one can presume of the model. I realize that this may be the default setting of the program, but it is hardly necessary. I miss a discussion of why the local blockage model may be inherently slower from a computational point of view than the wake model and the new global blockage model.
   **Response:**
   First of all, we highlight the need for accuracy in the convergence tolerance hyperparameter. In a setup with a large number of wind turbines and when taking the aggregate over a longer time interval, such as presented in this study, energy production can amount to an order of 1e5 GWh (see the results of A*). The convergence criterion defines for which update in wind speed (m/s) further iterations are no longer carried (see documentation: [https://topfarm.pages.windenergy.dtu.dk/PyWake/notebooks/EngineeringWindFarm Models.html](https://topfarm.pages.windenergy.dtu.dk/PyWake/notebooks/EngineeringWindFarmModels.html) ). Hence, since energy production is roughly proportional to the effective wind speed, an estimate on the level of one GWh requires a high accuracy in the convergence tolerance parameter.

   Furthermore, for large sites, an additional computational problem arises, which we discuss in our revised manuscript in Section 3.5. In a relatively regular layout with many interactions, the number of turbine interactions scales with $O(n^2)$, where $n$ denotes the number of turbines. Thus, the iterative nature in practice does lead to an inherently slower model, as the iterations must converge on all interaction combinations between turbines. This motivates the need for a different type of global blockage model which does not suffer from a superlinear increase in the number of turbines.

2. **Comment:** The coupling of the global blockage model utilizes some named features and methods in PyWake. The authors should rewrite these parts of the manuscript to make the methodology easier to follow for those reader not familiar with PyWake. Write down the appropriate equations explaining the coupling procedure and then

potentially include (maybe in an appendix) a description of how this can be accomplished in PyWake using already existing functionality.

**Response:** We thank Reviewer 2 for highlighting this. To further clarify the implementation structure, we have included a diagram highlighting the coupling setup in Section 3. Our implementation is not necessarily specific to PyWake, and can be used in all engineering wake model setups. . To clarify the coupling aspects in PyWake, however, we have included details in Section 3.1.1.

3. **Comment:** The new global blockage model depends on the height of the boundary layer H, the size of the wind farm (l) perpendicular to the flow direction, and a drag coefficient C_d. The model is built form analogy with blockage in a wind tunnel. The parameter C_d should depend on the geometry (porosity) of the wind farm. The authors choose a value of C_d=1 for simplicity. The wind farm size l is based solely on the front row. These choices may work OK for the regular layouts considered in the paper, but it is hard to trust that they will work equally for all types of layouts. The authors would do well to at least present an avenue for considering the details of the wind farm layout in future work.

**Response:** As motivated by this comment as well as comments of Reviewer 1, we have extended our experiment setup to also optimize the parameter C_d to LES data, providing an example of an implementation to model different layout geometries. Furthermore, we have added a discussion for a strategy of more extensive layout parametrization through LES studies for future work in the conclusions section.

4. **Comment:** The key parameter of the new model is the boundary layer height. This is known to have a significant influence on the wind farm flow. The authors use the mean value of H for four 90-degree wind direction sectors. Given that the LES data on which the work is based are inherently time series this seems like a peculiar choice. Why not run the calculations in a time series manner and fully leverage the dynamics of the boundary layer height? The authors make a point out the differences between performing a summation over directions before or after the errors are calculated. But for the important parameter H they are content with averaging over large wind direction sectors.

**Response:** We thank Reviewer 2 for raising this question on accuracy and granularity of H. The addition of running the wake model on time-series data is indeed a promising avenue for future work, which we have included Section 3.1 and the conclusions section. For the current paper however, we limit our scope to the steady-state wind flow modeling for which wake models have originally been designed. We agree with the concerns on the accuracy of the boundary layer height. Thus, in our revised version, we have adjusted our approach to take an atmospheric boundary layer height binned to every wind direction and speed bin in the wake model (averaged over 22.5 degrees wd and 1m/s ws), included in Section 4.2. We thank Reviewer 2 for this conceptual improvement.

5. **Comment:** Another nice addition would be to consider a less regular layout with a different wind rose to assess how well the calibrated model can be applied at a very

different site.

**Response:** We highlight that already in the case of our site for wind setup A*, the corner as seen from the south-west direction gives a much more irregular shape compared to the north-east direction which is used for model training purposes. Furthermore, since new wind directions are included with different wind characteristics compared to the training data (as can be seen in the wind rose), this in essence gives a view of how the calibrated model parameters carry over to another site. In view of the model assumptions, we deem it to be out of the scope of this research to include even more irregular (e.g. highly non-convex) layouts, although this is an interesting avenue for further research, which we have included in the conclusions section of our manuscript.

We now consider the numbered comments:

1. **Comment:** *What is the hub height and rotor diameter of the turbines? This information is needed if one is the replicate the results.*
   **Response**: this information is included in our citation of Gaertner et al. (2020), highlighting the technical specifications of the turbine. For additional clarity we have restated this in-text.

2. **Comment**: *How is the boundary layer height calculated from the LES data? Specifically, given the many different definitions of the boundary layer height what is the chosen definition in this work meant to signify physically and how is this related to the proposed global blockage model?*
   **Response**: As in the response to Reviewer 1, we have included a description in Section 4.2.

3. **Comment**: *How is the coupling between the wake model and the new global blockage model done? Is it only one-way or is it iterated?*
   **Response**: as highlighted in our response to Reviewer 1, we have presented a schematical overview of this (one-way) coupling in a diagram, included in Section 3 of the manuscript.

4. **Comment**: *What was the period for the LES data?*
   **Response**: we have included the LES data period (the calendar year 2023) in Section 4.2.

5. **Comment:** *Was any filtering applied on wind speeds or did all the cases include all wind speeds from the LES?*
   **Response:** Cases A, B, C include only northeastern wind speeds, whereas A* includes all wind speeds, as highlighted in our manuscript. For additional clarity, we have highlighted this again in Section 4.4.

6. **Comment**: *From Figure 2 it seems that the chosen local blockage model does not modify the flow field does not modify the flow field downstream of the rotor. However, in reality the flow is accelerated around the rotor. There are other local blockage models that include this effect.*
   **Response**: We thank Reviewer 2 for raising awareness of this "speedup" effect. In fact, this model (SelfSimilarityDeficit) does create a speedup acceleration effect (see e.g. https://topfarm.pages.windenergy.dtu.dk/PyWake/notebooks/BlockageDeficitModels.

html#SelfSimilarityDeficit). We note that the colorbar in our plot was "clipped off" for wind speeds > 10 m/s which hid this graphical result. We have updated the plot and included a mention of this effect in Section 3.

7. **Comment**: *Line 206: should northeastern wind speed be northeastern wind directions?*
   **Response**: We thank reviewer 2 for noticing this mistake and have corrected this in the manuscript.

8. **Comment**: *Figure 8: the grey dots are hard to see in the plot. Choose a more distinctive colour.*
   **Reponse**: We have updated the Figure and thank Reviewer 2 for raising this concern.

9. **Comment:** 2: *Line 253: are predictions errors the RMSE?*
   **Response**: These errors are indeed RMSE and we have highlighted this in the text

10. **Comment**: *Line 301: what does deficit inducing wake shape mean?*
    **Response:**  We have adjusted the phrasing used for clarity.

11. **Comment**: *The crescent features of the plots, see first panel near the point (14, 14), are so prominent that they call for an explanation. What are we seeing here?*
    **Response**: We have added an explanation and thank Reviewer 2 for pointing out this effect.

12. **Comment**: *Figure 11: the caption states that values for the A\*case appear more correlated because they are aggregated over all wind directions. It would be useful then to see the results from this case for isolated wind direction (sectors). Perhaps the four quarters of the unit circle.*
    **Response**: We have added this plot (see Figure 14) and thank Reviewer 2 for highlighting this.

13. **Comment**: *Figure 11: I assume there is one dot per turbine, but this should be made clear in the caption.*
    **Reponse:** we have highlighted this fact in the description for additional clarity.